# A study of the extraordinarily strong and tough silk produced by bagworms

Taiyo Yoshioka [1], Takuya Tsubota [2], Kohji Tashiro[3], Akiya Jouraku[4] & Tsunenori Kameda[1]

Global ecological damage has heightened the demand for silk as 'a structural material made from sustainable resources'. Scientists have earnestly searched for stronger and tougher silks. Bagworm silk might be a promising candidate considering its superior capacity to dangle a heavy weight, summed up by the weights of the larva and its house. However, detailed mechanical and structural studies on bagworm silks have been lacking. Herein, we show the superior potential of the silk produced by Japan's largest bagworm, *Eumeta variegata*. This bagworm silk is extraordinarily strong and tough, and its tensile deformation behaviour is quite elastic. The outstanding mechanical property is the result of a highly ordered hierarchical structure, which remains unchanged until fracture. Our findings demonstrate how the hierarchical structure of silk proteins plays an important role in the mechanical property of silk fibres.

[1] Silk Materials Research Unit, National Agriculture and Food Research Organization (NARO), 1-2 Owashi, Tsukuba, Ibaraki 305-8634, Japan. [2] Transgenic Silkworm Research Unit, National Agriculture and Food Research Organization (NARO), 1-2 Owashi, Tsukuba, Ibaraki 305-8634, Japan. [3] Department of Future Industry-Oriented Basic Science and Materials, Graduate School of Engineering, Toyota Technological Institute, Tempaku, Nagoya 468-8511, Japan. [4] Insect Genome Research and Engineering Unit, National Agriculture and Food Research Organization (NARO), 1-2 Owashi, Tsukuba, Ibaraki 305-8634, Japan. Correspondence and requests for materials should be addressed to T.K. (email: kamedat@affrc.go.jp)

Severe global concerns about growing ecological damage and depletion of non-renewable resources have heightened the demand for silk as a structural material made from renewable resources[1–3]. More than 200,000 different silks are known to exist in nature[4]. When we focus on toughness, i.e., a balance of strength and extensibility, we find the dragline silks of some kinds of spiders, such as those of the *Araneus*, *Nephila*, and *Latrodectus* genera, to be among the most attractive[5–7]. Particularly, the *Caerostris darwini* (*C. darwini*) (aka *Darwin's bark*) spider dragline silk had been considered the toughest silk in nature, showing five times higher toughness than that of the *Bombyx mori* (*B. mori*) silkworm silk[8] (see Table 1). However, many spider dragline silks, especially that of *C. darwini*, show typical plastic deformation in their tensile behaviours, showing a distinct yield point after the initial elastic region, followed by a subsequent levelling-off or plateau region before strain-hardening[8,9]. To realise their practical application as structural materials, stronger and tougher silks with more elastic deformation behaviours are required. Towards this goal, many trials are being conducted worldwide to produce ideal silk products, for instance, by controlling the hierarchical structure of regenerated silk proteins[10,11], producing transgenic (or genome-edited) artificial silk proteins[12–14], and searching for superior un-explored silks in nature[8,15,16]. When we search for superior un-explored silks, spiders provide a major hint that the silks used for dangling should be strong and tough[17–19]! Bagworm silk, produced by larvae of bagworm moths (*Lepidoptera* order and *Psychidae* family)[20] especially of the largest and heaviest Japanese bagworm *Eumeta variegata* (*E. variegata*) (a synonym for *Eumeta japonica* and *Clania variegata*)[21,22], might be a promising candidate, considering its superior capacity to dangle a heavy weight, summed up by the weights of the larva and its house (generally called larval case, bag, or nest). However, little is known about the mechanical and physical properties of bagworm silks[23–25].

In this report, based on the detailed analyses of the fibre morphology and mechanical properties, we reveal that the *E. variegata* bagworm silk is extraordinarily strong and tough compared to other known silks, and that its tensile deformation behaviour is quite elastic. To investigate the relationship between the structure and the outstanding mechanical properties of this bagworm silk, a comprehensive analysis of the hierarchical structure, made up of crystalline and amorphous phases, is conducted by in situ time-resolved simultaneous measurements of synchrotron wide-angle X-ray diffraction (WAXD) and small-angle X-ray scattering (SAXS) during tensile deformation.

## Results

**Morphology of bagworm silk**. While the exterior of the nest of *E. variegata* bagworms is covered with dead branches and leaves (Fig. 1a), the interior is made of fine, densely stacked nest silk fibres, like a nonwoven fabric (Fig. 1b, c). It is well-recognised that bagworms use their silk also as lifeline for dangling. Furthermore, we found that they use it as a foothold as well, by spinning it in a zigzag manner with slightly greater widths than that of the space between their legs, in which folded points are attached with adhesive glue. While spiders have plural pairs of silk glands and spinnerets for different purposes, the multi-task thread is produced by a pair of silk glands via a single spinneret in the case of bagworms. The individual silk threads are composed of a pair of thin filaments, as in the cases of silkworm silks and spider dragline silks, and their cross-sections were revealed to be rather elliptical, and not circular (Fig. 1d). The fact that *E. variegata* bagworms spin silk over several tenths of a metre to several hundreds of metres is worthy of mention; therefore, one can wound it on a bobbin (Fig. 1e) or twist it into multiple threads (Fig. 1f).

**Mechanical properties**. In the estimation of the tensile properties, the cross-sectional area of the sample filament should be evaluated as accurately as possible. Our detailed morphological observation revealed that the cross-sections were well-approximated by an ellipse, with a pair of major and minor axes $L_a$ and $L_b$, satisfying an axial ratio ($L_a/L_b$) of 1.7 (Fig. 1g–j; Table 2). This axial ratio was used to estimate the cross-sectional area of the single filaments, needed for converting the tensile force to stress values in the tensile test (further details in the Methods section).

A typical stress–strain curve of *E. variegata* bagworm silk is shown in Fig. 1k, and the average tensile properties of modulus, strength, extensibility, and toughness are summarised in Table 3. The values of modulus, strength, and toughness are extraordinarily high compared to other known silks (note that the toughness is almost comparable with that of *C. darwini* silk[8]). In addition, the *E. variegata* bagworm silk exhibits an ideal stress–strain behaviour, without a levelled-off stress after the yield point, but followed by a linear and steep strain-hardening just after elastic deformation.

**Amino acid sequence**. The structural and mechanical properties of the silk produced by *Lepidoptera* are predominantly attributable to the contribution of heavy-chain silk fibroin (H-Fib). An analysis of the amino acid composition, carried out for *E. variegata* bagworm silk, revealed that the molar ratios of Gly and Ala are approximately equal, and their sum accounts for about 80% (Supplementary Fig. 1), which is consistent with the results in earlier reports[26,27]. In accordance with this, we found, via silk gland transcriptomic analysis, that the bagworm *h-fib* gene hypothetically encodes the Gly and Ala–rich protein. The molecular structure of the H-Fib was investigated by a long-read transcriptomic analysis. At least 5 tandem repeat motifs were identified, each of which is composed of polyalanine block (PAB) and non-polyalanine block (NPAB) sequences (Fig. 2). This kind of combination motif of PAB and NPAB sequences is widely observed in the *Saturniidae* silkworm and spider dragline silks[28–30]. However, one can find several decisive distinctions from their motifs. Firstly, the length of each sequence motif,

**Table 1 Tensile properties of various silks**

|  |  | Young's modulus (GPa) | Fracture strength (GPa) | Extensibility | Toughness (MJ m$^{-3}$) | Ref. |
|---|---|---|---|---|---|---|
| *Araneus diadematus* | Spider silk (MA) | 10 | 1.1 | 0.27 | 160 | 54 |
| *Nephilla clavipes* | Spider silk (MA) | 13.8 | 1 | 0.20 | 111.2 | 61 |
| *Latrodectus hesperus* | Spider silk (MA) | 10.2 | 1 | 0.45 | 180.9 | 61 |
| *Caerostris darwini* | Spider silk (MA) | 11.5 | 1.7 (1.0)[a] | 0.52 (0.69)[b] | 354 | 8 |
| *Bombyx mori* | Silkworm silk | 7 | 0.6 | 0.18 | 70 | 54 |
| *Eumeta minuscula* | Bagworm silk | 25 | – | – | – | 24 |

[a]The true stress and [b]true strain values from the original paper were converted into engineering stress and strain and are given in parentheses

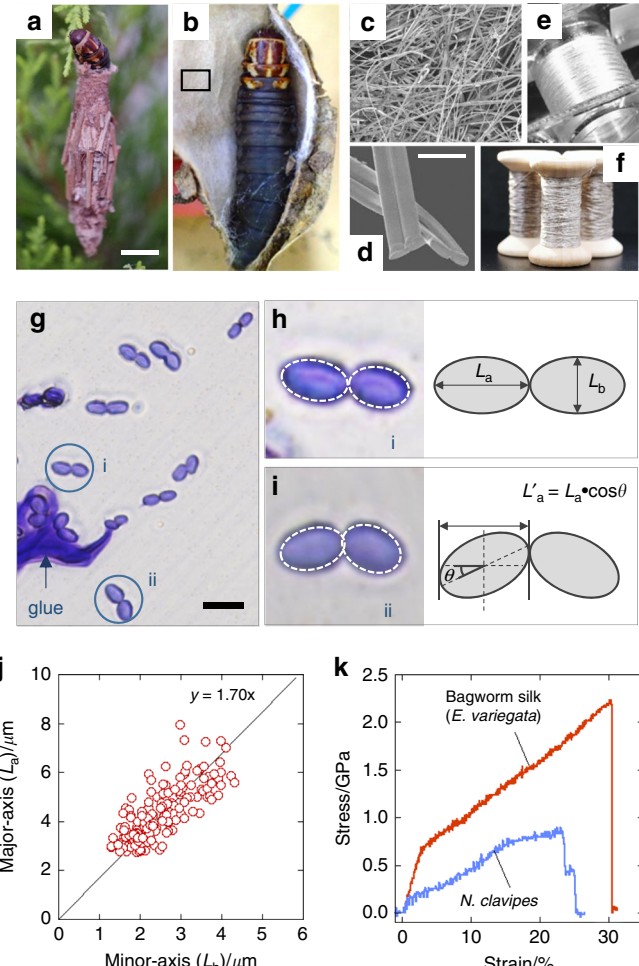

**Fig. 1** Morphological and mechanical characteristics of the bagworm silk. Photographs of **a** exterior and **b** interior of larval house (or nest) of a last-instar female larva of *E. variegata*. **c** SEM image of inner surface of the larval house (from rectangular part in **b**). **d** Magnified SEM image of single fibres with their cross-sections. **e** Continuous bagworm silk wound on a bobbin, and **f** its twisted yarn. **g** Optical micrograph of cross-sections of bagworm silks. Typical cross-sections of a single fibre composed of **h**, a pair of straight conjugated filaments, and **i** a pair of filaments conjugated with a tilting angle $\theta$, are shown together with schematic illustrations. In each filament in **h**, **i**, the cross-section is traced by an elliptical shape with a broken line. **j** Plot of axial ratios of lengths of major and minor axes in elliptical cross-sections of bagworm silk. The straight line with a slope of 1.70 was obtained by linear regression for the plots. **k** Typical stress–strain curve of single bagworm silks. For comparison, a typical stress–strain curve of *N. clavipes* dragline silk, measured in this study using the same procedure and instruments used for bagworm silks, is included. The average tensile properties of *N. clavipes* ($n = 5$) were as follows: modulus = 8.13 GPa, strength = 0.9 GPa, extensibility = 0.26, and toughness = 139.7 MJ m$^{-3}$. Scale bars; 10 mm (**a**), 10 μm (**d**), and 10 μm (**g**)

composed of ~160 amino acid units, is remarkably longer than the repeating units of ~20–40 known for the *Saturniidae* silkworm silks[28], and of ~30–60 for the spider dragline silks[29,30]. Secondly, the NPAB sequence is found to be made up of a combination of the long Gly-X dipeptide sequences (X is mainly Ala or Ser) observed characteristically in the *B. mori* silkworm silk[31], and the Gly-Gly-X tripeptide sequences (X is mainly Ala or Tyr) observed characteristically in the *Saturniidae* silkworm and spider dragline silks[28–30]. The long sequence of over 20 Ala

residues in the PAB is also very unique, compared to that of the *Saturniidae* silkworm (~3–15 repeats) and spider dragline silks (~5–8 repeats)[28–30]. It should be noticed that a short non-repeated sequence of ~5–8 residues, made up of the relatively bulky residues of mainly Ser, Val, and Tyr, is characteristically inserted in each NPAB sequence (coloured blue).

**Crystal modification and hierarchical structure.** To understand the structural origin of the outstanding mechanical properties of the bagworm silk, the crystal modification and hierarchical structure, which is a combination of crystal and amorphous phases, were investigated. The two-dimensional (2D) WAXD fibre diagram shows a typical $\beta$-sheet pattern[32] (Fig. 3a) widely observed in the silkworm and spider dragline silks. Although the $\beta$-sheet crystals formed in a variety of silks are commonly indexed with the orthogonal unit cell, the detailed unit cell parameters differ slightly among the silks depending on their amino acid composition[33,34]. The $\beta$-sheet unit cell parameters of the bagworm silk were evaluated, using the 200, 210, and 002 reflections, under the assumption of a rectangular unit cell, to be $a = 9.39$ Å, $b = 9.50$ Å, and $c$ (fibre axis) = $6.98 \pm 0.05$ Å, which are quite similar to those of the *B. mori* silk ($a = 9.38$ Å, $b = 9.49$ Å, and $c = 6.98$ Å)[32]. The crystallinity was estimated to be about 44% by peak fitting the crystalline and amorphous diffraction peaks (Fig. 3b).

The meridional 1D-SAXS Kratky profile ($q^2 I(q)$ vs. $q$ plot)[35] (Fig. 3d) (the original $I(q)$ vs. $q$ plot is shown in Supplementary Fig. 2) obtained from synchrotron SAXS measurement (Fig. 3c) clearly revealed at least five scattering peaks, the peak positions of which approximately satisfy an integer ratio of 1:2:3:4:5. This relation revealed the existence of a well-ordered repeating structure, composed of alternating crystal and amorphous phases. The first-order long period, that is, the real period ($L_p^{1st}$), was estimated to be about 38.5 nm. Analysis of the correlation function $K(z)$[36,37] (Fig. 3e) for the electron density distribution $\eta$ ($z$) of the repeating system made up of phases I and II (Fig. 3f) revealed that the 38.5 nm periodic structure consists of phase I with thickness of 15.7 nm and phase II with a thickness of 22.8 nm. The meridional periodic scatterings in SAXS are observed not only in this bagworm silk, but also in many kinds of spider dragline and wild silkworm silks. However, to the best of our knowledge, such a clear periodicity for fifth-order scattering has never been reported in any kind of $\beta$-sheet silk. In addition, the long period is roughly five times longer than those (~6–8 nm) reported for spider dragline silks and wild silkworm silks[38–43] (see Supplementary Fig. 3). This highly ordered and long-range hierarchical structure is considered to contribute significantly to the outstanding mechanical properties, and might be directly related to the amino acid sequence.

The SAXS data also revealed two-point scatterings on the equatorial line, indicating the presence of a nanofibril bundle with a neighbouring fibril distance of 4.7 nm (Fig. 3g). This periodicity is approximately the same as the periodicities observed for *B. mori* silkworm silk, *A. assama* wild-silkworm silk[38], and (*N. clavipes*) spider dragline silk[39] (Supplementary Fig. 3a, b). It is interesting that the fibre diameters of these silks considerably differ from each other, but the diameter of each minimum constituent nanofibril entity is similar. This is probably determined by the lateral spacing of the $\beta$-sheet crystallite[39]. The significantly sharper peak width of bagworm silk, compared to that of other silks, indicates a well-developed nanofibril bundle thickness. The nanofibril bundle thickness was estimated using the Scherrer equation and was around 150 nm, which is considerably thicker than those of other silks (Supplementary Fig. 3c).

**Table 2 Parameters of fibre morphology of *E. variegata* bagworm silk**

|  | Major axis ($L_a$)/$\mu$m | Minor axis ($L_b$)/$\mu$m | Axial ratio ($L_a/L_b$) | Tilt angle ($\theta$)/° |
|---|---|---|---|---|
| Bagworm silk (Last instar) | 4.42 (±0.66) | 2.55 (±0.48) | 1.73 | 10.37 (±7.50) |

Values in parentheses are ± standard deviation (±SD). The total sampling number was 185

**Table 3 Tensile properties of *E. variegata* bagworm silk**

|  | Young's modulus (GPa) | Fracture strength (GPa) | Extensibility | Toughness (MJ m$^{-3}$) |
|---|---|---|---|---|
| Bagworm silk | 28.1 (±2.1) | 2.0 (±0.2) | 0.32 (±0.03) | 364.0 (±44.1) |

Values in parentheses are ± standard error of the mean (±SEM). The total sample number n was 19 (Supplementary Fig. 6).

AAAAAAAAAEAAAAAAAAAAAAGSGAGAGGAGGYGAGAGAGAGAGAGGAAGAGGAGGAGGAGGYGGA
SVVYVGGGGAGAGAGSGAGAGSGAGAGAGSGAGAGGAGAAAGAGAGAGSGAGSGSGAGAGAGSGAGAG
SGAGAGAGAGSGAAGGAGAGAGAGAG

AAAAAAAAAEAAAAAAAAAAAAGSGAGAGGAGGYGAGAGAGAGAGAGGAGGAGGAGGAGGAGGYGGAG
VVYVSAGGAGAGAGSGAGAGSGAGAGAGSGAGAGGAGAAAGAGAGAGSGAGSGSGAGAGAGSGAGAGS
GAGAGAGAGSGAAGGAGAGAGAG

AAAAAAAAAEAAAAAAAAAAAAGSGAGAGGAGGYGAGAGAGAGAGAGGAGGAGGAGGAGGAGGYGGAG
VVYVSAGGAGAGAGSGAGAGSGAGAGAGSGAGAGGAGAAAGAGAGSGAGSGSGAGAGAGSGAVAGS
GAGSGAAGGAGAGAGAG

AAAAAAAAAEAAAAAAAAAAAAGSGAGAGGAGGYGAGAGAGAGAGAGGAGGAGGAGGAGGYGGA
SVVYVGAGGAGAGAGSGAGAGSGAGAGAGSGAGAGGAGAAAGAGAGAGSGAGSGSGAGAGAGSGAGAG
SGAGAGAGAGSGAAGGAGAGAG

AAAAAAAAAEAAAAAAAAAAAAGGAGGYGPYGGFAGAGAGAGGAGGAGGAGGAGGAG
STLIIVDEGGYGGAGGAGSGAGSGVGAGAGSGAGAGGAGAAAGAGAGAGSGAGSGSGAGAGAGSGAGA
GSGAGAGAGSGAAGGAGAGAGAG

**Fig. 2** Tandem repeat motifs in the bagworm H-Fib. Five tandem repeat motifs, each of which is composed of a PAB and NPAB sequence, were identified by silk gland transcriptomic analysis. The red characters indicate the polyalanine sequence and the blue characters show non-repeating residues, which are unique to each unit

**Structural changes during tensile deformation**. WAXD and SAXS analyses of the as-spun bagworm silk revealed a distinctive structural nature, that is, a highly ordered hierarchical structure made up of about 45% $\beta$-sheet crystalline phase and 55% amorphous phases. To clarify how the ordered structure contributes to the strength and toughness, we investigated the deformation behaviour of this highly ordered structure using in situ time-resolved simultaneous measurements of synchrotron WAXD, SAXS, and tensile deformation behaviour (the experimental setup is shown in Supplementary Fig. 4). The stress–strain plot is shown at the top of Fig. 4d, and the changes in the 2D-WAXD and SAXS patterns during tensile deformation are shown in Fig. 4a. Compared with the case of single fibres (Fig. 1k; Table 3), the fibre bundle (top of Fig. 4d) showed a non-negligibly lower fracture stress, and the stress decreased gradually in a broader strain region (stage III) due to the imperfection of parallelism of the bundle fibres. However, except for this difference in the stress–strain curve, the mechanical behaviour can be assumed to be essentially the same as that of a single fibre, allowing us to investigate the deformation process of the silk fibre via the in situ measurements performed for the bundle of fibres.

It was found that the lattice spacing of the meridional 002 crystal reflection ($d_{002}$) changed against the bulk strain in an approximately similar manner as that of the stress-strain curve behaviour (Fig. 4b, d). The once-deformed $d_{002}$ value recovered almost perfectly after fracturing (in Stage III). While the bulk strain shows an inflection point at a tensile stress of 0.4 GPa, the

crystal strain increased linearly against the tensile stress, until the fracture point (~1.4 GPa) (Fig. 5a). This is one of the characteristic differences from other kinds of silk, as will be discussed later in Fig. 5b, c, and is considered to be of crucial importance for understanding the structural origin of the strength and toughness of the bagworm silk, as discussed below.

The changes in the long periods $L_p^{2nd}$ and $L_p^{3rd}$ (Fig. 4c) are plotted against the bulk strain, in Fig. 4d, and the changes in the true long period ($L_p^{1st}$) were estimated from the changes in the $L_p^{3rd}$ to be from 37.5 to 44.7 ($\varepsilon_L = 19.2\%$) (Table 4). The ratio of the long periods ($L_p^{2nd}/L_p^{3rd}$) remained at a constant value of 1.5 throughout the tensile deformation (Fig. 4d). This demonstrated that the highly ordered repeating structure of crystal and amorphous phases was maintained throughout the stretching deformation. While the *c*-axis lattice spacing $d_{002}$ was elongated more pronouncedly in the initial elastic region (stage I) compared to the subsequent stage II after the yield point, the changes in the long period showed an opposite tendency. The elongated long period was not recovered, but remained constant after the tensile force was removed upon fracture. Similarly, the degree of crystal orientation was not recovered (Fig. 4d). These irreversible behaviours are essentially different from those observed for the dimensional change in the crystal unit cell.

## Discussion

We have revealed that the *E. variegata* bagworm silk is a promising candidate for sustainable structural materials, owing to its

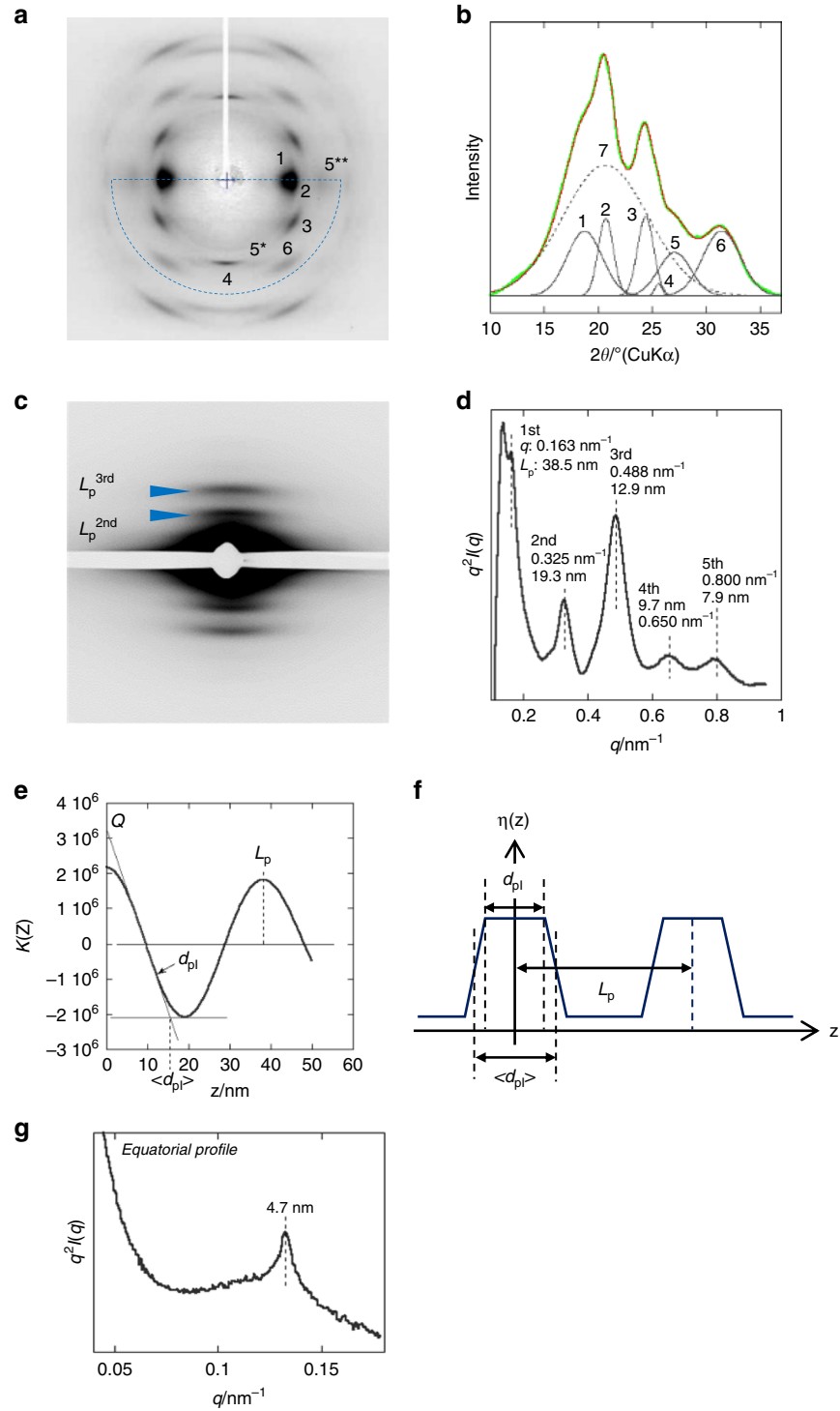

**Fig. 3** Structural information of the bagworm silk. **a** WAXD fibre diagram obtained from the bagworm silk and **b** its $2\theta$ profile scanned in the $2\theta$ range from 10° to 37°, and for the azimuthal angle range from 0° to −180°, which is enclosed by the broken blue line in **a**. The crystallinity was evaluated from the $2\theta$ profile by peak fitting. Each peak number corresponds to the following. 1: (200) crystalline reflection, 2: (210), 3: (211), 4: (002), 5: (102)* + (300)**, 6: (202), and 7: amorphous peak. The peak fitting was performed using the spectroscopy software Grams Suite 9.3 (Thermo Fisher Scientific Inc., USA) by fixing the peak positions for the crystalline reflections at ~1–6. **c** Synchrotron SAXS pattern and **d** its meridional $q$-profile ($q^2I(q)$ vs. $q$ Kratky plot). To avoid beam damage of the detector due to the strong equatorial-streak scattering, a narrow metal plate was attached on the detector surface along the equatorial line, and it appears as the white gap running along the equatorial line in the SAXS pattern **c**. **e** The electron density correlation function $K(z)$ and **f** the corresponding electron density distribution $\eta(z)$, for the repeating phases. $Q$, $L_p$, $d_{pl}$, and $\langle d_{pl} \rangle$ denote the invariant, long period (phase I + II), core thickness of phase I, and mean thickness of phase I, respectively. The $L_p$, $d_{pl}$, and $\langle d_{pl} \rangle$ were estimated to be 38.2, 12.5, and 15.7 nm, respectively. **g** Equatorial SAXS $q$-profile ($q^2I(q)$ vs. $q$ Kratky plot) scanned from the 2D pattern shown in Supplementary Fig. 3a-(1)

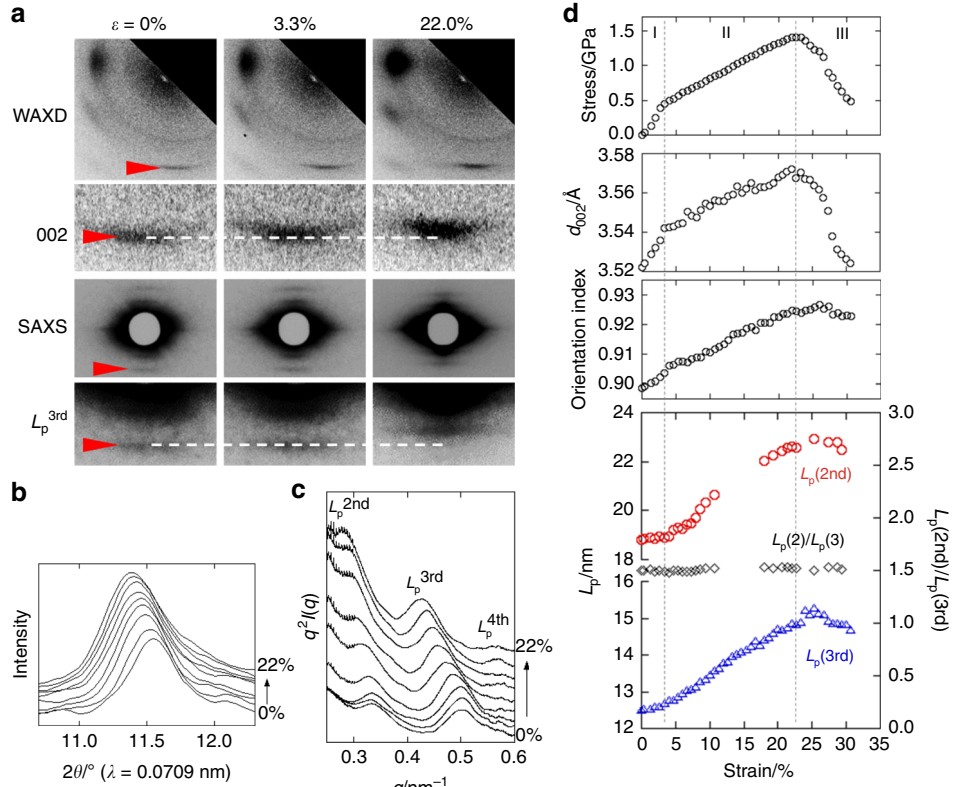

**Fig. 4** Time-resolved tracing of structural changes during tensile deformation. **a** Changes in time-resolved synchrotron WAXD and SAXS patterns during the tensile deformation process. The corresponding stress–strain curve is shown at the top of **d**. **b** The selective meridional WAXD $2\theta$ profiles and **c** the meridional SAXS $q^2 I(q)$ vs. $q$ profiles, both of which were obtained from the strains at 0, 1.3, 3.3, 6.0, 8.7, 12.0, 14.7, 18.0, and 22.0% in the stress–strain curve shown at the top of **d**. **d** Summary of the time-resolved simultaneous measurement. From top to bottom, the changes in the tensile stress, crystal unit cell dimension along the $c$-axis direction ($d_{002}$), crystalline orientation index, and meridional long periods ($L_p^{2nd}$ and $L_p^{3rd}$), estimated from the 2nd and 3rd periodic scatterings, are plotted against the bulk strain. In the bottom plots of the long periods, the ratio of $L_p^{2nd}/L_p^{3rd}$ is also plotted

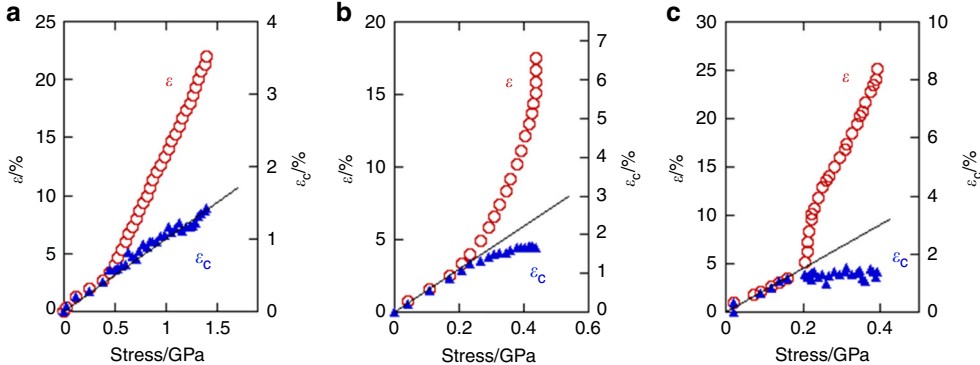

**Fig. 5** Relationships between crystal strain and tensile stress in different kinds of silk. The changes in bulk strain ($\varepsilon$) and of crystal strain ($\varepsilon_c$) against the tensile stress in the time-resolved simultaneous synchrotron X-ray analysis for **a** *E. variegata* bagworm silk, **b** *B. mori* (*Bombycidae*) silkworm silk, and **c** *A. assama* (*Saturniidae*) silkworm silk. The changes in tensile stress and crystal strain against the bulk strain, using the same experimental data, are shown in Supplementary Fig. 5

**Table 4 Summary of the dimensional changes observed in the bulk sample, crystal unit cell, and hierarchical structure**

|  | Stage I (Initial elastic region) | Stage II (Strain hardening) | Total (I + II) | Stage III (After fracturing) |
|---|---|---|---|---|
| Sample length (mm) | 15.0 → 15.5 | 15.5 → 18.3 | 15.0 → 18.3 | not evaluated |
| Bulk strain ($\varepsilon$) (%) | 3.3 | 18.7 | 22.0 | not evaluated |
| $d_{002}$ (Å) | 3.52 → 3.54 | 3.54 → 3.57 | 3.52 → 3.57 | 3.57 → 3.52 |
| Crystal strain ($\varepsilon_c$) (%) | 0.57 | 0.85 | 1.41 | −1.40 |
| Long period ($L_p^{1st}$) (nm) | 37.5 → 38.1 | 38.1 → 44.7 | 37.5 → 44.7 | 44.7 → 44.0 |
| Strain of long period ($\varepsilon_L$) (%) | 1.6 | 17.6 | 19.2 | −1.6 |

superior mechanical properties and their quite elastic tensile-deformation behaviour. While the β-sheet crystal modification revealed by WAXD analysis is common in the other kinds of silkworm and spider silks, the hierarchical structure was found to be quite unique. The SAXS analysis revealed a highly ordered and long-range repeating structure. Additionally, while the *B. mori* silk does not show the meridional long period, many *Saturniidae* silkworm and spider dragline silks show it. In other words, such ordered structure is considered to be constructed only in the silks having a PAB sequence, and the PAB plays an important role in the formation of the periodic structure. The PAB of the *Saturniidae* silkworm or spider dragline silks is considered to form a helical conformation in their silk glands[44,45]. It is reasonably speculated that the polyalanine helices in the neighbouring H-Fib proteins (or spidroins) are gathered around each other, probably forming a hexagonal packing via hydrophobic–hydrophobic self-assembly interactions[10,46,47]. The long PAB sequence of 22 residues (corresponding to 6 helical pitches) in the bagworm silk (Fig. 6a, b) may gather together more effectively compared to the shorter alanine repeats of the *Saturniidae* and spider dragline

silks. This self-assembly of the helical polyalanine sequences occurs at approximately constant intervals determined by the amino acid residues of the tandem repeat motifs (~160 residues in the case of the *E. variegata* bagworm silk) (Fig. 6c), and these intervals are speculated to correspond to the long period detected by SAXS measurement (Fig. 6d). Here, the question is: how many amino acid sequence units contribute to the β-sheet crystal formation from the end of the PAB sequence? We paid attention to the presence of the non-repeated short sequences inserted in each NPAB sequence region. It was found that around 45 residues exist between the PAB and non-repeated short unit. The 67 residues (22 in PAB and 45 in NPAB) were calculated to measure 23.5 nm (~0.35 nm/residue in the β-sheet crystal[32]), which is consistent with the length of phase II (22.8 nm) experimentally estimated by SAXS analysis (Fig. 6d). We made an assumption, which is widely believed, that the polyalanine helical conformation changes to β-sheet crystals due to the stress-induced helix-sheet transition[10,47]. We can assign the lengths of phases I and II to the amorphous and crystalline phases, respectively (Supplementary Note 1). The remaining NPAB sequence of around 90 residues should behave

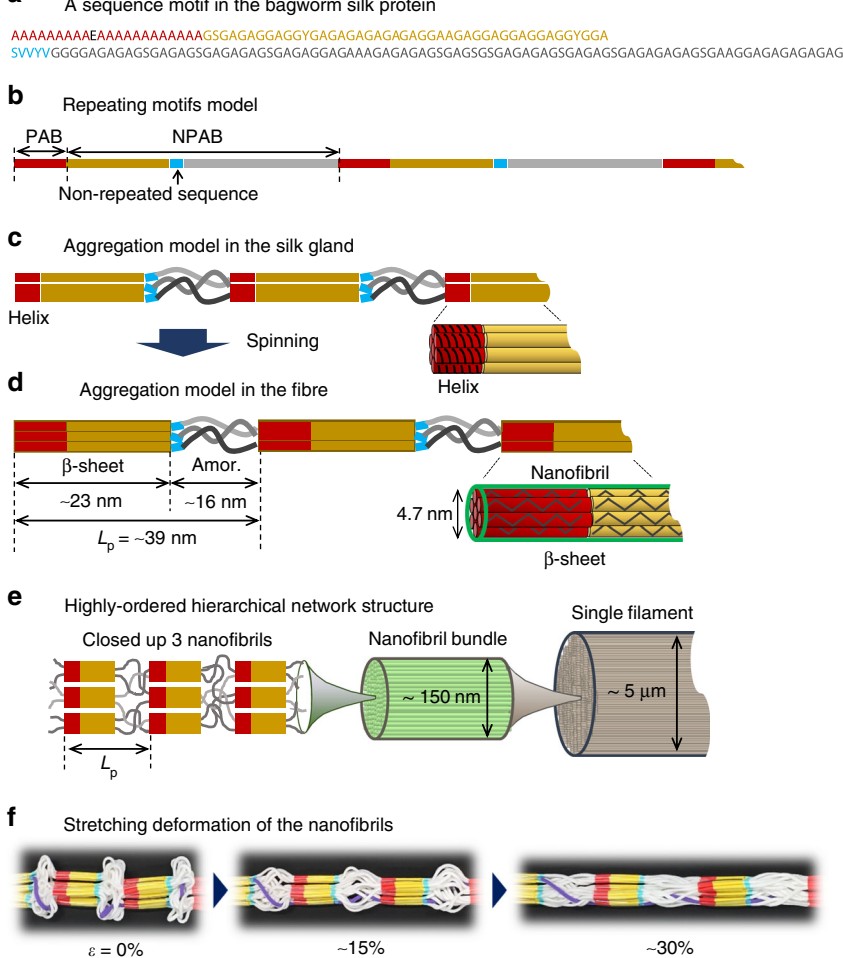

**Fig. 6** Relationship between primary structure and hierarchical structure. **a** A representative sequence motif, corresponding to the top of Fig. 2, identified in the H-Fib protein of the *E. variegata* bagworm silk, and **b** the repeating model of the motifs. **c** Assumed aggregation model of the motifs in the silk gland, and **d** after spinning. In the silk gland, the PAB has a helical conformation, and the neighbouring helical PABs gather together to form the aggregation bundle (or nanofibril) with 4.7 nm thickness and hexagonal packing, as shown in **c**. During the spinning process, the neighbouring NPAB sequences between the PAB and blue-coloured non-repeated sequences crystallise with β-sheet modification as shown in **d**. Simultaneously, the helical PAB bundle exhibited a stress-induced structural transition from helix to β-sheet crystals (as shown in **d**). SAXS analysis revealed a well-developed nanofibril bundle with a thickness of around 150 nm for a single filament, as shown in **e**. A stretching deformation model of the nanofibrils, during which the ordered repeating structure is maintained, is proposed in **f**. In situ X-ray analysis revealed that the deformations of the crystal and amorphous phases are reversible and irreversible modes, respectively

as amorphous, by shrinking about 50% from the $\beta$-sheet conformation $((90 \times 0.35 - 15.7)/(90 \times 0.35) \times 100\ (\%))$ (picture on the left in Fig. 6d).

The tensile deformation behaviours of the different silks differ significantly among the species. The in situ time-resolved synchrotron X-ray measurement during the tensile deformation of the bagworm silk successfully revealed a remarkable structural feature, wherein the crystal strain increased linearly against the tensile stress throughout the stretching, up to the fracture point (Fig. 5a). For comparison, we conducted the same measurements for the *B. mori* and *A. assama* silks, which yielded different types of stress–strain curves. All the three kinds of silk initially show a linear increase in the crystal strain for a low tensile stress, as reported by Seydel et al. for the *B. mori* silk[48], but saturated at a later stage in the *B. mori* and *A. assama* silks (Fig. 5b, c, and Supplementary Fig. 5). This indicates that the stress distribution to the crystal phases in the bagworm silk is more homogeneous throughout the tensile process than that in the other two silks. This superior stress distribution mode should be realised owing to the continuous and highly ordered periodic structure revealed by SAXS analysis. By combining all the experimental results obtained in the present study, we have proposed a hierarchical structure model for bagworm silks (Fig. 6e). The ellipsoidal single filament is made up of the 150 nm-thick bundles of nanofibrils, each of which has a diameter of about 4.7 nm. These nanofibrils are considered to be assembled together and form the repeatedly-arranged structure with the amorphous phases. It is reasonably speculated that the amorphous chain segments gather together randomly and the entire system of nanofibrils appears like a network of nanofibrils (image on the left in Fig. 6e). The tension-induced structural deformation of the nanofibril bundle is schematically illustrated in Fig. 6f, where the ordered repeating structure is maintained in this process. Our findings obtained by X-ray scattering techniques are consistent with the intrinsic nanofibrillar nature of silkworm and spider silks revealed by nanoscale imaging techniques[49,50]. Verification of the proposed structural model and the deformation process of bagworm silks using nanoscale imaging and spectroscopy techniques remains an avenue for further study.

Our experimental results presented an important fact: the mechanical property of the crystal phase of silk $\beta$-sheets is considerably higher than what we had considered so far. The slope of the stress *vs.* crystal strain plot of the bagworm silk gave a crystal modulus of 106 GPa (Fig. 5a), which is four times higher than the values for *B. mori* silkworm silks estimated by the X-ray diffraction method[51,52]. The methodology for the measurement of the crystal modulus is established, but it is made under an assumption of homogeneous stress distribution (Supplementary Note 2). The high modulus of the bagworm silk proposed a possibility that the true crystal modulus of the $\beta$-sheet crystal is much higher than what we had considered[53], and that the stress distribution to the crystal phases is far from an ideal homogeneous stress distribution, but quite heterogeneous, in natural silks (Supplementary Note 3). In contrast, the highly ordered network structure of bagworm silk results in a highly homogeneous stress distribution mode and gives the crystal modulus approximately close to the true ultimate modulus.

In summary, we found that the bagworm silk is extraordinarily strong and tough compared to other known natural silks, and its tensile deformation behaviour is quite elastic. Although a variety of silks with different mechanical properties should be candidates for various practical applications, bagworm silk seems especially appropriate for use as a structural material. The detailed structural analyses, from amino acid sequence to hierarchical structure level, clarified that the outstanding mechanical properties originated from a highly ordered hierarchical network structure,

which remains unchanged up to fracture in the tensile deformation process. Such unique structural and mechanical functions are largely attributed to the requirements of protecting and supporting the larva, where the former requirement is crucial for silkworm cocoon silks and the latter for spider dragline silks. Silkworm cocoon silks have evolved to protect the larva during their pupal stage, and therefore stiffness is a crucial aspect[18]. On the other hand, spider dragline silks, the web-frame silk, have evolved to bring rapidly flying prey as well as to support the spider's body as her lifeline. Therefore, spider dragline silks are tougher, which enables absorption of the kinetic energy of the prey or the spider itself without breaking[18,54,55] (the architectures of spider webs are also sophisticated to the requirements[55,56]). The tandem repeat motifs of the H-Fib protein of bagworm silk show an ideal sequence composition to produce this hybrid silk from a pair of glands, combining the functions of silkworm and spider dragline silks. Our findings clearly demonstrate how the hierarchical structure of silk proteins plays an important role in the mechanical properties of the resultant bulk fibres, and how it is closely related to the primary structure of the amino acid sequence. We believe that the knowledge obtained through this will lead to fruitful strategies for designing the primary structures, resulting in stronger and tougher artificial silks, based on genome-editing technology[57]. It should be emphasised here that the strong potential of bagworm silks is not only owing to the superior mechanical properties, but also the promising possibilities for practical applications. We have successfully developed a mass rearing technique and fibre collection method for bagworms, with a continuous and long scale over several hundreds of metres, without sacrificing the animals (Fig. 1e, f). This strong, tough, and long silk thread is anticipated to be applicable to various kinds of material fields, such as modern apparel, and biomedical and opto-electrical fields, and an especially promising candidate for the next generation of structural materials made from renewable resources, conforming to the spirit of animal welfare.

## Methods

**Silk fibre collection**. *E. variegata* bagworm larvae of the last instar, within a size distribution reported by Sugimoto[21], were collected in Tsukuba City, Ibaraki (Japan) in October 2015. Single fibres for the tensile test were collected from the wall of the rearing cage on which bagworms spun their silk as footholds. The fibre bundles for X-ray analysis and for time-resolved simultaneous measurement of X-ray diffraction and tensile tests were also collected from the wall of the rearing cage. Once stress is loaded, the initial long period irreversibly changes, as revealed in this study; therefore, all the sample fibres were carefully corrected without loading excessive tension.

**Observation of surface morphology**. Observations of the fibre surface morphology were made using a digital camera (Stylus Tough, Olympus Co., Japan), fluorescence microscope in optical microscopy mode (BZ-X700, Keyence Co., Japan), and scanning electron microscope (SEM) (JSM-6301F, JEOL Ltd., Japan).

**Observation of cross-sections**. Cross-sections of the bagworm silks were prepared as described below and examined using the BZ-X700 microscope in optical mode. The silk fibres were gently aligned in a parallel fashion and embedded in an epoxy resin (Quetol 812, Nisshin EM Co. Ltd., Japan). Thereafter, the resin was polymerised at 60 °C for 48 h and the samples were sliced across the fibre axis to thicknesses ranging from 1.0 to 2.0 μm using an ultra-microtome (LKB 2088, LKB Produkter, Sweden). The resultant thin sections were stained with a 1% (w/v) aqueous solution of toluidine blue containing 1% (w/v) borax.

**Evaluation of cross-sectional area**. While most of the fibres are made up of a pair of conjugated filaments with an almost linear relation between their major axes as schematically illustrated in Fig. 1h, for some fibres, the major axes of the pair of filaments were found to be conjugated at an angle of $180 - 2\theta$ (°), where the angle $\theta$ is defined as the tilt angle of each major axis from the linear relation, as schematically illustrated in Fig. 1i. The $\theta$ for each fibre was evaluated using the Image J open-source software to be approximately 10.37°.

The cross-sectional area of each fibre was estimated as follows. Firstly, the cross-sectional shape of each thin filament was assumed to be a perfect ellipse with an

axial ratio ($L_a/L_b$) of 1.7, as described in the main text. By this approximation, one can calculate the cross-sectional area by measuring the length of the major axis of the filament using a well-calibrated BZ-X700 instrument. At this stage, the $\theta$ should be taken into account, because the length of the major axis measured using the optical microscope ($L_a'$) is $L_a \cdot \cos\theta$, as illustrated schematically in Fig. 1i. Therefore, the actual cross-sectional area of a single filament is $\frac{\pi}{4 \times 1.70 \times \cos^2\theta} \cdot L_a'^2$. The average tilt angle of 10.37° was used as the value of $\theta$.

**Tensile test**. The tensile properties of the individual fibres of *E. variegata* bagworm silk were measured at 22–25 °C and 40–60% relative humidity, using a mechanical tensile stage (EZ Test/CE, Shimadzu Co., Japan) equipped with a 5 N load cell. All the measurements were carried out at a cross-head speed of 10 mm min⁻¹. A single fibre made up of a pair of thin filaments with an elliptical cross section was attached to a handmade paper flame using epoxy glue, with a window distance (i.e., sample distance) of 15 mm. After positioning the paper flame holding a single fibre in the tensile stage, both sides of the flame were cut gently, and then the measurement was started. The cross-sectional area of each fibre was estimated for every sample, based on optical microscope observations. The procedure for estimating the elliptical cross-sectional area of bagworm silks was described in the previous section. In all, 19 measurements (4 or 5 measurements each for 4 different bagworms) were carried out and averaged (Supplementary Fig. 6).

**Analysis of amino acid composition**. The amino acid composition of the bagworm silk fibres was analysed with a Shimadzu amino acid analysis system using a high-performance liquid chromatography CBM-20A controller (Shimadzu Co., Japan), equipped with the columns Shim-pack Amino-Na (100 mm (L) × 6.0 mm ($\phi$)) and ISC-30 (Na) (50 mm (L) × 4.0 mm ($\phi$)) (Shimadzu Co., Japan). The de-gummed fibres were dissolved in 6 N hydrochloric acid under vacuum at 110 °C for 22 h. The sample was dried under nitrogen gas purging, re-dissolved in pH 2.2 sodium citrate buffer, and examined, for the amino acid analysis.

**RNA sequencing and bioinformatics analysis**. The larvae of the bagworm moth were collected in Abiko City, Chiba (Japan) in September 2015. The silk gland was dissected from one larva and the total RNA was extracted from the posterior region of the silk gland using the ISOGEN (NIPPON GENE, Tokyo, Japan) and SV Total RNA Isolation System (Promega, Madison, WI). The long-read sequencing analysis (Isoform Sequencing) was carried out using a PacBio RS II sequencer. The library was prepared using the Clontech SMARTer PCR cDNA Synthesis Kit (TaKaRa, Kusatsu, Japan), and the ~5–10 kb libraries were selected using the BluePippin Size-Selection System (NIPPON GENETICS, Tokyo, Japan). After purification and end-repair, the blunt-end SMARTbell adaptors were ligated. The libraries were quantified using a Quant-IT PicoGreen kit (Thermo Fisher Scientific, Waltham, MA), and qualified using the Agilent Technologies 2100 Bioanalyzer (Agilent Technologies, Waldbronn, Germany). Subsequently, the libraries were sequenced using the PacBio P6C4 chemistry in an 8-well SMART Cell v3 with a PacBio RS II system. For sequence correction, the Illumina short-read sequence was used. The total RNA extracted from the posterior region of the silk gland was applied for the RNA library construction using a TruSeq RNA Sample Prep Kit v2 (Illumina, San Diego, CA), and the 101 bp paired-end reads were obtained using an Illumina HiSeq 2500 sequencer. Adaptor sequences and low quality bases in the Illumina short reads were trimmed using the Trimmomatic version 0.32[58], and the cleaned short reads were used for the long-read correction with CoLoRMap[59]. The raw sequence data have been deposited in the DNA Data Bank of Japan (DDBJ) Centre (https://www.ddbj.nig.ac.jp/index-e.html) and can be accessed via accession numbers, DRA007344 (short-reads) and DRA007345 (long-reads).

**Static structural analysis by WAXD and SAXS**. A parallel-aligned fibre bundle with a thickness of about 0.1 mm was prepared, and the surface adhesive was washed with a 0.05 M sodium carbonate boiled aqueous solution for 3 min The fibre bundle was then thoroughly rinsed and dried, and transferred as samples for the subsequent X-ray analyses.

For references, the de-gummed *B. mori* and Indian *Antheraea assama* (*A. assama*) (aka *Muga* silk) (*Saturniidae*) silks were also gently collected in a parallel-aligned fibre bundle with a bundle thickness of about 0.5 mm, and used for the subsequent X-ray analyses during the tensile deformation test. The cross-sectional area for each fibre bundle, needed for the data conversion from the force-strain to the stress-strain curves, was estimated from the length and weight of the initial fibre bundle and the density for each silk. The values of density, 1.35 and 1.31 g cm⁻³, reported by Gupta et al. for *B. mori* and *A. assama* silks, respectively, were used[60].

The crystalline modification of the bagworm silk was investigated by WAXD analysis, using an R-Axis Rapid II X-ray diffractometer (Mo-Kα) (Rigaku Co., Japan) equipped with a cylindrical-type imaging plate camera. The hierarchical structure was investigated by SAXS analysis, using a synchrotron X-ray beam in the SPring-8 40B2 beamline or using a NANOPIX 3.5 M X-ray diffractometer (Cu-Kα) (Rigaku Co., Japan) equipped with the highly sensitive single-photon-counting pixel detector HyPix-6000 (Rigaku Co., Japan). Cerium oxide and silver behenate were used to calibrate the camera distances for the WAXD and SAXS

measurements, respectively. All X-ray measurements were performed at 22–25 °C and 40–60% relative humidity.

**Time-resolved synchrotron X-ray scattering investigation**. The structural changes in the bagworm silk, occurring in the stretching processes, were investigated by time-resolved simultaneous measurements of the force-strain curve and synchrotron WAXD and SAXS patterns. The fibre bundle with a bundle thickness of about 0.1 mm was set on a stretching device, the Micro-stretcher (Linkam Scientific Instruments Ltd., UK), which was set in a SPring-8 40B2 synchrotron X-ray beamline (Hyogo, Japan). The setup geometry is shown in Supplementary Fig. 4. The WAXD and SAXS data were measured every 9 s (8 s exposure time +1 s interval) during the continuous stretching process, with a stretching rate of 10 μm s⁻¹. The wavelength of the incident X-ray beam was 0.0709 nm, and the sample-to-camera distances for the WAXD and SAXS measurements were 95 and 763 mm, respectively. The beam size at the sample position was around 0.4 × 0.8 mm². Tuning of the best beam intensity, which yields the best quality of scattering without serious sample damage during the repetitive measurements, was carefully performed using the attenuator system. A flat panel detector with a pixel size of 50 × 50 μm² (Hamamatsu Photonics K. K., Japan) and CMOS detector with a pixel size of 43.1 × 43.1 μm² (Hamamatsu Photonics K. K., Japan) were used for the WAXD and SAXS measurements, respectively. For the accurate calibration of the WAXD camera distance, silicon powder was dusted onto the fibre bundle. The force-strain data obtained from the tensile test was approximately converted to a stress-strain curve on the basis of the stress-strain curves obtained from the single fibres. The sample strain (%) was estimated as $\Delta l$ (mm)/$l_0$ (mm) × 100 (%), where $l_0$ is the initial sample length of 15 mm between the tensile clamps, and $\Delta l$ is the stretched length.

**WAXD data analysis**. For all the cases of WAXD analysis, air scattering was subtracted from the original 2D-WAXD pattern. The crystallinity was estimated from the WAXD $2\theta$ profile. The $2\theta$ profile was scanned from the 2D-WAXD fibre diagram, and ranged from 10 to 37°. For the $2\theta$ profile obtained, peak fitting analysis was performed to separate the contributions from the crystalline reflections and amorphous halo-scattering. For the peak fitting, the contributions from the 6 crystalline lattice planes, (200), (210), (211), (002), (102) + (300), and (202), and that from the single amorphous scattering, were considered. Each $2\theta$ position of the crystalline reflections was calculated on the basis of the unit cell parameters of the $\beta$-sheet crystal determined for the bagworm silk in this study. The peaks were fitted with Gaussian functions, using a "non-linear least squares fitting", in the spectroscopy software Grams Suite 9.3 (Thermo Fisher Scientific Inc., USA). The crystallinity $X_c$ was calculated using Eq. (1):

$$X_c(\%) = \frac{I_c}{(I_c + I_a)} \times 100, \tag{1}$$

where $I_c$ and $I_a$ are the total intensities from the crystal and amorphous phases, respectively.

The crystal strain ($\varepsilon_c$) along the $c$-axis direction at each bulk strain ($\varepsilon$) was evaluated from the changes in the $2\theta$ position of the 002 meridional reflection. The $\varepsilon_c$ was defined as shown in Eq. (2).

$$\varepsilon_c(\%) = (d_{002}(\varepsilon) - d_{002}(\text{initial}))/d_{002}(\text{initial}) \times 100 \tag{2}$$

The crystalline orientation index was evaluated using the definition (180-FWHM)/180, where FWHM is the full width at half maximum of the azimuthal angular distribution of the WAXD equatorial 200 reflection.

**SAXS data analysis**. For all the cases of SAXS analysis, air scattering was subtracted from the original 2D-SAXS pattern. The meridional or equatorial $q$-profiles were scanned from the meridional and equatorial lines with an azimuthal angle range of ±10°, where $q$ is the scattering vector, defined as $(4\pi/\lambda)\sin\theta$, where $\lambda$ is the wavelength of the X-ray beam used. Thus, the obtained 1d-$q$-profile ($I(q)$ vs. $q$ plot) was replotted as a Kratky plot ($q^2I(q)$ vs. $q$ plot) to enhance the peak features more evidently[35].

The one-dimensional electron density correlation function $K(z)$ between a periodic structure made of phases I and II, which gives an electron density distribution $\eta$(z), as depicted in Fig. 3f, was calculated from the SAXS data. The correlation function $K(z)$ is defined as shown in Eq. (3)[36].

$$K(z) \propto \int_0^\infty q^2 I(q) \cos(zq)\,\mathrm{d}q \tag{3}$$

Q, $L_p$, $d_{pI}$, and $\langle d_{pI}\rangle$, shown in Fig. 3e, f, denote the invariant, long period, core thickness of domain I, and mean thickness of phase I, respectively.

The strain ($\varepsilon_L$) in the long period was estimated using Eq. (4).

$$\varepsilon_L(\%) = (L_p^{1st}(\varepsilon) - L_p^{1st}(\text{initial}))/L_p^{1st}(\text{initial}) \times 100 \tag{4}$$

The thickness of the nanofibril bundle was evaluated from the broadening of the equatorial SAXS peak, using the Scherrer Eq. (5)[10]:

$$d = \frac{K\lambda}{\beta \cos\theta}, \tag{5}$$

where $d$ is the thickness of the nanofibre bundle, $\lambda$ is the wavelength of the incident X-ray beam (CuK$\alpha$ 0.15418 nm radiation), $\theta$ is the angle of the considered reflection, $\beta$ is the line broadening at half-maximum intensity of the considered reflection, and $K$ is a dimension-less shape factor (Scherrer factor). For the shape factor $K$, a value of 0.9 was used. The $\beta$ was corrected for the instrumental broadening, which was estimated by measuring the powder diffraction profile of silver behenate, on the basis of the Eq. (6).

$$\beta = \sqrt{\beta_{obs}^2 - \beta_{AgBeh}^2} \tag{6}$$

## Data availability

The raw sequence data of the next generation sequencer have been deposited in the DNA Data Bank of Japan (DDBJ) Centre (https://www.ddbj.nig.ac.jp/index-e.html) and can be accessed via accession numbers, DRA007344 (short-reads), and DRA007345 (long-reads). The same raw sequence data is available from the Open Science Framework: https://doi.org/10.17605/OSF.IO/3GYSU. Data supporting the findings of this study are available from the authors upon reasonable request.

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

## Acknowledgements
This research was partly supported by Science and Technology Research Partnership for Sustainable Development (SATREPS), Japan Science and Technology Agency (JST) / Japan International Cooperation Agency (JICA) (T.K.). K.T. thanks MEXT, Japan for their support through the "Strategic Project to Support the Formation of Research Bases at Private Universities (2015–2019)". The synchrotron radiation experiments were performed using the SPring-8 BL40B2 with the approval of Japan Synchrotron Radiation Research Institute (JASRI) (Proposal Nos 2015B1184 and 2016A1440) (T.Y.). The authors thank Dr. N. Ohta (JASRI) for his professional support in the synchrotron X-ray experiments, and Ms. F. Yukuhiro of the National Agriculture and Food Research Organisation (NARO) for the sample preparation related to the cross-sections. The authors also thank Dr. S. Niitsu and Mr. O. Saito for providing the bagworm moth.

## Author contributions
All the authors (T.Y., T.T., K.T., A.J., and T.K.) contributed equally to the work and manuscript preparation.

## Additional information

**Competing interests:** Patent applications have been filed for the findings described in this publication (PCT/JP2017/023839 and PCT/JP2017/037327). While the research was done in an academic/institutional environment, since then the material has been taken up by a company to develop for commercialisation.

