## [Transparent Peer Review File · Nature Communications]

Reviewers' comments:

Reviewer #1 (Remarks to the Author):

This very interesting paper reports a quite intriguing discovery of superior mechanical properties of a Lepidopteran silk. Lepidoptera larvae were the first to be domesticated for "industrial" production of silks and, for some reason, the community of materials scientists shifted its attention to spider silks ignoring the second largest order (~160 000 Lepidoptera species) of insects. I am not surprised, that lepidopterans with such a rich biology, overperform other fiber spinning arthropods. This paper unveils the superior materials properties of their silks exemplified by *E. variegata*. It is a new discovery opening up new avenues for research and is very attractive from the practical standpoint as one can rely on the oldest textile manufacturing methodology to scale up production of this silk. This manuscript warrants publication in Nature Communications after addressing minor comments.

1. I am not sure why the biology of Lepidoptera and Araneae silks should be comparable. The authors do discuss the common protein sequences and differences between these silks, however, I did not find anything related to the organismal biology and spinning habits of these arthropods. It would be interesting to comment on the rate of filament formation in these arthropods. Is anything known about the water content in the prefiber of bagworms?

2. Fig. 5 is informative and important. It might be a good idea to add some inserts showing a hypothetical deformation of the fiber pictured in Fig. 6 at each stage of loading. This will help the reader to appreciate the importance of the model sketched in Fig. 6

3. Schematics c) - e) in Fig.6 should be redrawn. In the text, the authors talk about a lamellar structure, however, in Fig.6, no lamellae are shown. It takes a lot of time to decipher the specified periods, what do they really mean. I do not understand how the beta-sheets are formed: in the figure it says Zigzag, but I do not see how can a hexagonally packed crystal deform in a zigzag. Do the authors assume that the beta-sheet forms a scroll?

4. I believe that the measurements and interpretation of the X-ray diffraction pattern are correct. What is missing is a clear and simple schematic explaining the author's hypothesis about deformation of lamellar structure and amorphous regions. While the interpretation is given in the text, it is very difficult to draw a simple visual picture of what is said about irreversible deformation in the fiber and bulk crystals. This leaves the proposed mechanism of deformation hidden.

5. Such a schematic is also necessary to support the statement, lines 205-209: "While the bulk strain shows an inflection point at a tensile stress of 0.4 GPa, the crystal strain increased linearly against the tensile stress, until the fracture point (~1.4 GPa) (Fig. 5a). This is one of the characteristic differences from the other kinds of silk, and is considered to be of crucial importance for understanding the structural origin of the strength and toughness of the bagworm silk".

In summary, I found this manuscript very interesting and exciting, the results are novel and have the potential to change the direction in research on bioinspired mechanically superior fibers.

Reviewer #2 (Remarks to the Author):

This is an excellent article supporting the claim of the authors made in the title. The contents of the article are of interest to a large audience.

The authors were fortunate observing much better defined lamellar X-ray scattering for *E. Variegata* than for spider dragline (major ampullate) silk fibres or other silkworm fibres allowing

them performing a state-of-the-art SAXS analysis. The data support the nanofibrillar nature of silks, contrasting with some AFM data interpretations (1). SAXS/WAXS provides, however, only bulk hierarchical microstructural information. In view of the discussion of nanostructural features - such as skin/core structures in spider dragline silks (2)- it would be interesting studying bagworm silk fibres by nanoscale imaging and spectroscopy techniques.

I have a few suggestions for improving the text, some of which are more of interest to specialists and should be included in the Supporting Information.

Introduction: A comparison of the mechanical properties -such as toughness- for spider and silkworm silks has limitations as spider dragline silk is optimized for rapidly dissipating the energy of an object impacting the orb-web (3) while bagworm or silkworm silks are obviously used for constructing and supporting larva and houses in a more static way. One could argue that this is at the origin of the lower crystallinity of spider dragline silk (about 12%)(4) as compared to bagworm silk (about 44%). In addition, set-aside "dangling" applications, spider dragline silk and the much more flexible flagelliform silk are the main fibres used for orb-web construction. The combination of both silks is required for the superb energy dissipation properties of orb-webs which might find practical technological applications (5). The authors should therefore discuss in more detail the different functional aspects of these silks.

Line 63 Young's modulus of *T. ephemeraeformis* (6) is much lower than that of *E. Variegata*. X-ray diffraction pattern reveals also a lower crystallinity.(6) Do the authors have any information on functional differences which could explain this?

Lines 82/83 "Unlike that of spiders, the multi-task thread..." The authors should formulate this more carefully as the "structural, multi-task" major ampullate silk thread of spiders is spun from only two spinnerets while other spider silks (e.g. aciniform) are spun from multiple spinnerets.

Lines 105/106 Provide a brief comparison with a few other technological fibres such as aramid, either in the text or in a table.

Line 175 The comparison of fibril distances should be improved as the authors provide only selective information. Indeed, the value of 4.5 nm for dry dragline silk (7) has been recorded for an extremely long X-ray exposure which does not allow excluding radiation damage. Results on air-dry dragline fibres provide rather 6-7 nm values (8,9). The origin of the sample and experimental conditions for the curve in S.I. Fig.3B is not explained.

Line 208 Provide a reference for crystal strain increase for "other kinds of silks"

Figure 6 The authors use the term "hierarchical network structure" in the context of Fig.6 and the mechanical deformation. Fig.6 shows the hierarchical organization of semicrystalline nanofibrils into bundles and filaments but no 3D network with lateral interactions. I understand that around 67 residues of the molecular sequences are attributed to the nanofibrils and a further 90 residues are considered as amorphous suggesting a 3-phase model also discussed for semicrystalline polymers. The elastic properties of dragline silk are usually discussed in terms of an amorphous network reinforced by nanodomains which implies again a 3-phase model if one considers the nanodomains as being part of the nanofibrils. It might therefore be useful clarifying the term "network structure".

Supporting Information: Provide more details on:

- Wavelength, flux, beam size at the sample position, type of detector and calibration of detector distance.
- Sample environment (temperature, humidity control?)
- How was radiation damage assessed & minimized, in particular for multiple exposures during

deformation experiments?

- Synchronization of diffraction and deformation experiments; step-wise strain increase or continuous?
- Estimation of errors of unit cell parameters (lines 149/150); which peaks were used to derive these.
- Information on the in situ deformation setup (e.g. reference).

References

- 1 Perez-Rigueiro, J., Elices, M., Plaza, G.R., & Guinea, G.V., Similarities and Differences in the Supramolecular Organization of Silkworm and Spider Silk. *Macromolecules* 40, 5360-5365 (2007).
- 2 Frische, S., Maunsbach, A., & Vollrath, F., Elongate Cavities and Skin-Core Structure in Nephila Spider Silk Observed by Electron Microscopy. *J. Microscopy* 189, 64-70 (1998).
- 3 Gosline, J.M., DeMont, M.E., & Denny, M.W., The Structure and Properties of Spider Silk. *Endeavour* 10 (1), 37-43 (1986).
- 4 Grubb, D.T. & Jelinski, L.W., Fiber Morphology of Spider Silk: The Effects of Tensile Deformation. *Macromolecules* 30 (10), 2860-2867 (1997).
- 5 Qin, Z., Compton, B.G., Lewis, J.A., & Buehler, M.J., Structural Optimization of 3D-Printed Synthetic Spider Webs for High Strength. *Nat Comm* 6, 7038 (2015).
- 6 Reddy, N. & Yang, Y., Structure and Properties of Ultrafine Silk Fibers Produced by *Theriodopteryx Ephemeraeformis*. *J. Mat. Sci.* 45, 6617-6622 (2010).
- 7 Miller, L.D. & Eby, R.K., A 45 Å equatorial long period in dry dragline silk of *Nephila clavipes*. *Polymer* 41, 3487-3490 (2000).
- 8 Riekkel, C., Burghammer, M., Ferrero, C., Dane, T., & Rosenthal, M., Nanoscale Structural Features in Spider Bridge Thread Fibres. *Biomacromolecules* 18 (1), 231-241 (2017).
- 9 Yang, Z., Grubb, D.T., & Jelinski, L.W., Small-Angle X-ray Scattering of Spider Dragline Silk. *Macromolecules* 30, 8254 - 8261 (1997).

Reviewer #3 (Remarks to the Author):

This study discusses the discovery of remarkable biomechanical properties, and structural details of bagworm silk. These silk fibers are among the strongest and toughest biomaterials known, and the ability to mass rear the animals may render these fibers more readily feasible for practical applications than those, for example, of spider silks that cannot be harvested in mass with ease. The study contains detailed structural as well as genetic characterization of the silk and thus goes beyond most similar studies. This paper is very well researched and presents exciting findings. There are some issues, however, that need to be addressed.

First, the English requires careful review by a native speaker. Spelling and grammar errors are many and sentence structure is often awkward. One of very many examples of sentence structure:

The tensile deformation behaviours of different silks are not similar, but differ significantly, depending on the species.

Should read something like "The tensile deformation behaviours of different silks differ significantly among species."

Second, there seems to be some discrepancy between the text and table on the strength of these fibers. In the text: "While the bulk strain shows an inflection point at a tensile 206 stress of 0.4 GPa, the crystal strain increased linearly against the tensile stress, until the fracture point (~1.4 GPa) (Fig. 5a)." This sentence seems at odds with the claimed strength of 2.0 GPa in table 3. Some spider silks have a fracture point higher than 1.4 GPa (e.g. Darwin's bark spider at 1.7 GPa),

and thus potentially exceed this species in strength.

Third, I think the findings are overstated. Apart from the issue mentioned above, the claim that this is the toughest silk found to date seems unwarranted. Average toughness of Darwin's bark spider samples from ref 7 is 354 MJ/m³ while the average reported here is 364 with standard error of 44. Is 364 +/- 44 significantly different from 354? What was the range of values? The toughest single fiber of Darwin's bark spider silk measured to date was 520 MJ/m³. Does any single bagworm silk fiber exceed this value? If either of these answers are no, then the claim "owing to its superior mechanical properties, that exceed any of all known silks" is overstated (this sentence is also awkward in structure). I'd say the toughness of these fibers are par with the toughest material previously reported.

Fourth, it is not clear why: "To realise their practical application as structural materials, stronger and tougher silks with more elastic deformation behaviours are required." This requires an explanation. It seems that a variety of silks with different biomechanical properties are excellent candidates for practical application. Greater elasticity may be good for some purposes and bad for others – don't generalize

Sincerely, Ingi Agnarsson

Referee 1

No	Referee comment	Answer	Modified part
	This very interesting paper reports a quite intriguing discovery of superior mechanical properties of a Lepidopteran silk. Lepidoptera larvae were the first to be domesticated for "industrial" production of silks and, for some reason, the community of materials scientists shifted its attention to spider silks ignoring the second largest order (~160 000 Lepidoptera species) of insects. I am not surprised, that lepidopterans with such a rich biology, overperform other fiber spinning arthropods. This paper unveils the superior materials properties of their silks exemplified by E. variegata. It is a new discovery opening up new avenues for research and is very attractive from the practical standpoint as one can rely on the oldest textile manufacturing methodology to scale up production of this silk. This manuscript warrants publication in Nature Communications after addressing minor comments.	Thank you for your valuable and insightful suggestions on our manuscript. Our detailed responses to your comments are listed below.	
1	I am not sure why the biology of Lepidoptera and Araneae silks should be comparable. The authors do discuss the common protein sequences and differences between these silks, however, I did not find anything related to the organismal biology and spinning habits of these arthropods. It would be interesting to comment on the rate of filament formation in these arthropods. Is anything known about the water content in the prefiber of bagworms?	We agree with your observation, that detailed comparisons as to the organismal biology and spinning habits are necessary for discussing the differences in the mechanical properties of silks produced by different kinds of insects. In our manuscript, a detailed discussion to this effect by comparing the water content in the pre-fibre stage, has not been provided. However, we have added a discussion regarding the differences in the structural and mechanical functions between the bagworm, silkworm, and spider dragline silks, based on the differences in the crucial aspects required for these silks. A detailed comparison among them, including the differences in organismal biology and spinning habits, remains a very important future topic.	Lines 322-333
2	Fig. 5 is informative and important. It might be a good idea to add	As per your suggestion, a schematic model of the tensile	Fig. 6f

	some inserts showing a hypothetical deformation of the fiber pictured in Fig. 6 at each stage of loading. This will help the reader to appreciate the importance of the model sketched in Fig. 6	deformation process of nanofibril bundle has been inserted as Fig. 6f.	Lines 295-297 Lines 813-816
3	Schematics c) - e) in Fig.6 should be redrawn. In the text, the authors talk about a lamellar structure, however, in Fig.6, no lamellae are shown. It takes a lot of time to decipher the specified periods, what do they really mean. I do not understand how the beta-sheets are formed: in the figure it says Zigzag, but I do not see how can a hexagonally packed crystal deform in a zigzag. Do the authors assume that the beta-sheet forms a scroll?	We understand that this comment is due to our inappropriate usage of the terminologies ‘lamellar structure’ and ‘zigzag’. In the original version, we used ‘lamellar structure’ to explain the repeating structure of the crystal phase and amorphous phase, which corresponds to the repetition of the 23 nm beta-sheet and 16 nm amorphous phases in Fig. 6d. We have now corrected ‘lamellar structure’ to ‘repeating structure of crystal and amorphous phases’. On the other hand, we used ‘zigzag’ to explain the molecular conformation of the polyaniline part in the beta-sheet crystal (thus, this does not mean a ‘zigzag beta-sheet’). Therefore, we have deleted the terminology ‘zigzag’ from the manuscript. The schematics c) ~ e) are important to relate the amino acid sequence and the aggregation state of the chain segments. We would like to retain these schematics.	(lamellar structure) e.g. line 163 (zigzag) e.g. Fig. 6d
4	I believe that the measurements and interpretation of the X-ray diffraction pattern are correct. What is missing is a clear and simple schematic explaining the author's hypothesis about deformation of lamellar structure and amorphous regions. While the interpretation is given in the text, it is very difficult to draw a simple visual picture of what is said about irreversible deformation in the fiber and bulk crystals. This leaves the proposed mechanism of deformation hidden.	As mentioned above (in response 3), our choice of the terminology ‘lamellar structure’ was not appropriate. We have now modified this to ‘repeating structure of crystal and amorphous phases’. The schematic explaining the deformation process of the repeating crystal and amorphous phases has been added to Fig. 6f.	Fig. 6f Lines 295-297 Lines 813-816
5	Such a schematic is also necessary to support the statement, lines 205-209: "While the bulk strain shows an inflection point at a tensile stress of 0.4 GPa, the crystal strain increased linearly against the tensile stress, until the fracture point (~1.4 GPa) (Fig. 5a). This is one of the characteristic differences from the other kinds of silk, and is considered to be of crucial importance for understanding the structural origin of the strength and toughness of the bagworm silk".	The detailed phenomenon occurring at the inflection point has not been clarified. Therefore, here, we can only show experimental results, which reveal a clear difference in the tensile deformation behaviours between the bagworm silk and other kinds of silks (Figs. 5b and c).	

Referee 2

No	Referee comment	Answer	Modified part
	This is an excellent article supporting the claim of the authors made in the title. The contents of the article are of interest to a large audience.	Thank you for your valuable and insightful suggestions on our manuscript. Our detailed responses to your comments are listed below.	
1	The authors were fortunate observing much better defined lamellar X-ray scattering for E. Variegata than for spider dragline (major ampullate) silk fibres or other silkworm fibres allowing them performing a state-of-the-art SAXS analysis. The data support the nanofibrillar nature of silks, contrasting with some AFM data interpretations (1). SAXS/WAXS provides, however, only bulk hierarchical microstructural information. In view of the discussion of nanostructural features -such as skin/core structures in spider dragline silks (2)- it would be interesting studying bagworm silk fibres by nanoscale imaging and spectroscopy techniques. I have a few suggestions for improving the text, some of which are more of interest to specialists and should be included in the Supporting Information. References : 1 Perez-Rigueiro, J., Elices, M., Plaza, G.R., & Guinea, G.V., Similarities and Differences in the Supramolecular Organization of Silkworm and Spider Silk. Macromolecules 40, 5360-5365 (2007). 2 Frische, S., Maunsbach, A., & Vollrath, F., Elongate Cavities and Skin-Core Structure in Nephila Spider Silk Observed by Electron Microscopy. J. Microscopy 189, 64-70 (1998).	Indeed, we believe that our X-ray scattering experiments support the intrinsic fibrillar nature of silkworm and spider silks, revealed by nanoscale imaging techniques so far. We will verify our proposed nanofibrillar model using such imaging and spectroscopy techniques in future work. We have briefly discussed this aspect in the revised manuscript.	Lines 297-301
2	Introduction: A comparison of the mechanical properties -such as toughness- for spider and silkworm silks has limitations as spider	We fully agree with your observation. In the revised manuscript we have discussed the differences in the structural	Lines 322-333

	dragline silk is optimized for rapidly dissipating the energy of an object impacting the orb-web (3) while bagworm or silkworm silks are obviously used for constructing and supporting larva and houses in a more static way. One could argue that this is at the origin of the lower crystallinity of spider dragline silk (about 12%)(4) as compared to bagworm silk (about 44%). In addition, set-aside “dangling” applications, spider dragline silk and the much more flexible flagelliform silk are the main fibres used for orb-web construction. The combination of both silks is required for the superb energy dissipation properties of orb-webs which might find practical technological applications (5). The authors should therefore discuss in more detail the different functional aspects of these silks. References: 3 Gosline, J.M., DeMont, M.E., & Denny, M.W., The Structure and Properties of Spider Silk. Endeavour 10 (1), 37-43 (1986). 4 Grubb, D.T. & Jelinski, L.W., Fiber Morphology of Spider Silk: The Effects of Tensile Deformation. Macromolecules 30 (10), 2860-2867 (1997). 5 Qin, Z., Compton, B.G., Lewis, J.A., & Buehler, M.J., Structural Optimization of 3D-Printed Synthetic Spider Webs for High Strength. Nat Comm 6, 7038 (2015).	and mechanical functions between bagworm, silkworm, and spider dragline silks, based on the differences in the crucial aspects required for these silks. We also expect that other silk research groups will constructively contribute in terms of the different functional aspects between the silks produced by bagworms, silkworms, and spiders.	
3	Line 63 Young’s modulus of T. ephemeraeformis (6) is much lower than that of E. Variegata. X-ray diffraction pattern reveals also a lower crystallinity. (6) Do the authors have any information on functional differences which could explain this? References: 6 Reddy, N. & Yang, Y., Structure and Properties of Ultrafine Silk Fibers Produced by Theriodopteryx Ephemeraeformis.	In our preliminary experiments, we performed WAXD measurement for 5 kinds of different Japanese bagworm silks; however, there was no such low crystallinity silk. Therefore, at this point, we are unable to provide a reasonable explanation as to why the crystallinity of T. ephemeraeformis is quite low.	
4	Lines 82/83 “Unlike that of spiders, the multi-task thread...” The authors should formulate this more carefully as the “structural,	As per your suggestion, we have carefully modified the related sentence.	Lines 82-84

	multi-task” major ampullate silk thread of spiders is spun from only two spinnerets while other spider silks (e.g. aciniform) are spun from multiple spinnerets.		
5	Lines 105/106 Provide a brief comparison with a few other technological fibres such as aramid, either in the text or in a table.	In this study, our main aims were 1) to determine the mechanical properties of bagworm silk and clarify the relationship with the hierarchical structure, and 2) to compare the mechanical properties between bagworm silk and other natural silks. Hence, at this stage, it would not appropriate to compare the mechanical properties of bagworm silk with those of synthetic fibres. Needless to say, this is an interesting topic for future work, in addition to comparison of their hierarchical structures.	
6	Line 175 The comparison of fibril distances should be improved as the authors provide only selective information. Indeed, the value of 4.5 nm for dry dragline silk (7) has been recorded for an extremely long X-ray exposure which does not allow excluding radiation damage. Results on air-dry dragline fibres provide rather 6-7 nm values (8,9). The origin of the sample and experimental conditions for the curve in S.I. Fig.3B is not explained. References: 7 Miller, L.D. & Eby, R.K., A 45 A equatorial long period in dry dragline silk of Nephila clavipes. Polymer 41, 3487-3490 (2000). 8 Riekkel, C., Burghammer, M., Ferrero, C., Dane, T., & Rosenthal, M., Nanoscale Structural Features in Spider Bridge Thread Fibres. Biomacromolecules 18 (1), 231-241 (2017). 9 Yang, Z., Grubb, D.T., & Jelinski, L.W., Small-Angle X-ray Scattering of Spider Dragline Silk. Macromolecules 30, 8254 - 8261 (1997).	Indeed, the hierarchical structure of silks is affected by the sample environment, especially by the humidity and temperature. Our measurements were performed at temperatures ranging from 22°C to 25°C and relative humidities of 40 % to 60 %. We have added these measurement conditions to the Methods section.	Lines 458-459
7	Line 208 Provide a reference for crystal strain increase for “other	The term ‘other kinds of silks’ indicates B. mori silk and A.	Line 215

	kinds of silks”	assama silk, shown in Fig. 5. To clarify this aspect, we have accordingly modified the related sentence.	
8	Figure 6 The authors use the term “hierarchical network structure” in the context of Fig.6 and the mechanical deformation. Fig.6 shows the hierarchical organization of semicrystalline nanofibrils into bundles and filaments but no 3D network with lateral interactions. I understand that around 67 residues of the molecular sequences are attributed to the nanofibrils and a further 90 residues are considered as amorphous suggesting a 3-phase model also discussed for semicrystalline polymers. The elastic properties of dragline silk are usually discussed in terms of an amorphous network reinforced by nanodomains which implies again a 3-phase model if one considers the nanodomains as being part of the nanofibrils. It might therefore be useful clarifying the term “network structure”.	To clarify the network structure, we have modified Fig.6e and the related sentences.	Fig. 6e Lines 288-295
9	Supporting Information: Provide more details on:  •Wavelength, flux, beam size at the sample position, type of detector and calibration of detector distance. •Sample environment (temperature, humidity control?) •How was radiation damage assessed & minimized, in particular for multiple exposures during deformation experiments? •Synchronization of diffraction and deformation experiments; step-wise strain increase or continuous? •Estimation of errors of unit cell parameters (lines 149/150); which peaks were used to derive these. •Information on the in situ deformation setup (e.g. reference). 	As per your suggestion, we have provided the following information.  1. Wavelength: ⇒ originally provided (lines 452, 455, and 470) 2. Flux: ⇒ Unfortunately, we do not have information about the flux. 3. Beam size (at the sample position): ⇒ newly added (lines 471-472). 4. Type of detector: ⇒ originally provided (lines 453, 456, and 474-477) 5. Calibration of camera distance: 	

		⇒ newly added (lines 457-458, 477-478) 6. Sample environment ⇒ newly added (lines 458-459) 7. Radiation damage minimization: ⇒ newly added (lines 472-474) 8. Experimental condition for deformation: ⇒ newly added (line 469) 9. Errors of unit cell parameters (and used reflections): ⇒ newly added (lines 151-153) 10. In situ deformation set up: ⇒ newly added (Supplementary Fig. S4) and (line 467)	
--	--	---	--

Referee 3

No	Referee comment	Answer	Modified part
	This study discusses the discovery of remarkable biomechanical properties, and structural details of bagworm silk. These silk fibers are among the strongest and toughest biomaterials known, and the ability to mass rear the animals may render these fibers more readily feasible for practical applications than those, for example, of spider silks that cannot be harvested in mass with ease. The study contains detailed structural as well as genetic characterization of the silk and thus goes beyond most similar studies. This paper is very well researched and presents exciting findings. There are some issues, however, that need to be addressed.	Thank you for your valuable and insightful suggestions on our manuscript. Our detailed responses to your comments are listed below.	
1	First, the English requires careful review by a native speaker. Spelling and grammar errors are many and sentence structure is often awkward. One of very many examples of sentence structure: The tensile deformation behaviours of different silks are not similar, but differ significantly, depending on the species. Should read something like “The tensile deformation behaviours of different silks differ significantly among species.”	Prior to submission, our manuscript was checked for language by native English speaker from editing service, and this time again for this resubmission. We have submitted the editing certificates to the journal editor. Nevertheless, as per your suggestion, we have rechecked the language and sentence structure of the manuscript to best of our abilities.	
2	Second, there seems to be some discrepancy between the text and table on the strength of these fibers. In the text: “While the bulk strain shows an inflection point at a tensile 206 stress of 0.4 GPa, the crystal strain increased linearly against the tensile stress, until the fracture point (~1.4 GPa) (Fig. 5a).” This sentence seems at odds with the claimed strength of 2.0 GPa in table 3. Some spider silks have a fracture point higher than 1.4 GPa (e.g. Darwin’s bark	Indeed, the mechanical property of the fibre bundle is poorer than of the single fibre. This may be due to the lack of perfect parallelism of the bundle fibres. However, we do not consider this a serious problem for our purpose of the in situ experiment, because their stress-strain behaviours are essentially the same as those of single fibres. Nevertheless, in order to clarify this aspect, we have added some sentences to the manuscript.	Lines 199-205

	spider at 1.7 GPa), and thus potentially exceed this species in strength.		
3	Third, I think the findings are overstated. Apart from the issue mentioned above, the claim that this is the toughest silk found to date seems unwarranted. Average toughness of Darwin's bark spider samples from ref 7 is 354 MJ/m ³ while the average reported here is 364 with standard error of 44. Is 364 +/- 44 significantly different from 354? What was the range of values? The toughest single fiber of Darwin's bark spider silk measured to date was 520 MJ/m ³ . Does any single bagworm silk fiber exceed this value? If either of these answers are no, then the claim "owing to its superior mechanical properties, that exceed any of all known silks" is overstated (this sentence is also awkward in structure). I'd say the toughness of these fibers are par with the toughest material previously reported.	Indeed, the toughness of bagworm silk is quite similar to that of Darwin's bark. This fact has been added to the revised manuscript. Furthermore, we have added all the stress-strain curves and mechanical properties measured in this study as Supplementary Fig. 6.	Lines 108-109 S.I.Fig. 6
4	Fourth, it is not clear why: "To realise their practical application as structural materials, stronger and tougher silks with more elastic deformation behaviours are required." This requires an explanation. It seems that a variety of silks with different biomechanical properties are excellent candidates for practical application. Greater elasticity may be good for some purposes and bad for others – don't generalize	We agree with your point, that a variety of silks with different biomechanical properties are excellent candidates for practical application. We have partly modified the related sentence to reflect this aspect.	Lines 316-319

REVIEWERS' COMMENTS:

Reviewer #1 (Remarks to the Author):

I am satisfied with the revised manuscript. All comments have been addressed. I like the schematic and it does help to understand the author's ideas. Change of terminology is also good. This manuscript warrants publication in the present form.

Reviewer #2 (Remarks to the Author):

The authors have adequately taken care of my comments to the first draft in the revised article. I particularly agree to the clarification of the term "lamellar structure" in response to remarks by referee 1. The hierarchical model is fully explained in Fig.6a-f and the "nanofibrils bundle" model in Fig.6e can be well related to the equatorial profile and the nanofibrils bundle thickness in S.I.Fig. 3B/C. The exciting research performed by the authors is clearly presented in the article and should provide the starting point for a fresh look on silkworm-type fibers. I support therefore publication of the article in Nature Communications.

Editorial Note: Given the lack of technical questions raised by reviewer #3 in the previous round of review and the author's responses, the editor decided not to go back to reviewer #3 for this round of review.